# Nanocomposite of MgFe_2_O_4_ and Mn_3_O_4_ as Polyphenol Oxidase Mimic for Sensing of Polyphenols

**DOI:** 10.3390/bios12060428

**Published:** 2022-06-17

**Authors:** Harmilan Kaur, Manpreet Kaur, Renuka Aggarwal, Sucheta Sharma, Davinder Singh

**Affiliations:** 1Department of Chemistry, Punjab Agricultural University, Ludhiana 141004, India; harmilan2020-cm@pau.edu; 2Department of Food and Nutrition, Punjab Agricultural University, Ludhiana 141004, India; renukaaggarwal@pau.edu; 3Department of Biochemistry, Punjab Agricultural University, Ludhiana 141004, India; suchetasharma_pau@pau.edu; 4Department of Extension Education, Punjab Agricultural University, Ludhiana 141004, India; davinder-ee@pau.edu

**Keywords:** polyphenol oxidase mimic, catechol sensing, resorcinol sensing, MgFe_2_O_4_@Mn_3_O_4_ NCs

## Abstract

Polyphenol oxidase (PPO) mimics have advantage of detection and remediation of polyphenols. This work demonstrates rapid and sensitive colorimetric detection of phenolic compounds using nanocomposite of magnesium ferrite (MgFe_2_O_4_) and manganese oxide (Mn_3_O_4_) nanoparticles as PPO mimic. The catalytic properties of MgFe_2_O_4_ and Mn_3_O_4_ displayed synergistic effect in the nanocomposite. The synthesized nanocomposite and nanoparticles were fully characterized using various analytical techniques. The ratio of MgFe_2_O_4_ and Mn_3_O_4_ in the nanocomposite was optimized. Catechol and resorcinol were taken as model polyphenols. The best PPO-activity was shown by MgFe_2_O_4_@Mn_3_O_4_ nanocomposite with of *w*/*w* ratio 1:2. The results correlated with its higher surface area. Reaction parameters viz. pH, temperature, contact time, substrate concentration, and nanoparticles dose were studied. The synthesized MgFe_2_O_4_@Mn_3_O_4_ nanocomposite was used for the detection of catechol in the linear range of 0.1–0.8 mM with the detection limit of 0.20 mM, and resorcinol in the range of 0.01–0.08 mM with the detection limit of 0.03 mM. The estimated total phenolic content of green and black tea correlated well with the conventional method. These results authenticate promising future potential of MgFe_2_O_4_@Mn_3_O_4_ nanocomposite as PPO-mimic

## 1. Introduction

The term “enzyme mimic” refers to a synthetic enzyme that performs similar functions as its natural analogue. Natural enzymes have higher catalytic efficiency, are substrate specific and are widely employed in the food processing sectors. However, they are prone to temperature and pH changes, which causes denaturation [1]. Enzyme mimics are being studied to circumvent these limitations of natural enzymes. Nanoparticles (NPs) are tiny particles having diameter ranging from 1 nm to 100 nm [2]. They differ from the bulk material in terms of catalytic properties and have a wide range of biological and catalytic applications [3]. Nanozymes, or NPs with enzyme-like properties, have been studied for their potential applications in biosensors, pharmaceutical and food sectors [4]. Nanozymes come in a variety of shapes and sizes, including rods, fibers, wires, tubes, and nanopopcorns, all of which have inherent capabilities to operate as enzyme mimics [5,6,7]. Magnesium ferrite (MgFe_2_O_4_) is a stable material with catalytic and magnetic properties. The flower-like MoS_2_@MgFe_2_O_4_ composites have been reported to simulate peroxidase activity [8,9]. Polyphenol oxidase mimics have immense potential for detecting phenolic substances such as catechol and resorcinol [10,11,12,13]. Due to their stability, toxicity, and carcinogenicity, phenols have been identified as hazardous water pollutant [14]. As a result, monitoring phenols in the environment is an important aspect of research. Although analysis methods, viz chemiluminescence and fluorometry have been developed for the detection of phenols with high sensitivity, they all have the inherent shortcoming of qualitative identification for phenols [15,16,17]. Electrochemical methods for the determination of phenolic compounds (e.g., catechin) too are preferred over other methods because of their fast response time, low detection limit and relatively low cost [18,19,20]. Chromatography and mass spectrometry have been used to identify and detect phenols with high accuracy and sensitivity, but they require expensive instruments and time-consuming operations, making them unsuitable for rapid analysis. Colorimetric approach, in comparison to other outlined methods, do not necessitate considerable instrumentation or preprocessing, and is also less expensive. It also offers the benefit of being able to analyze the data using visual detection [21]. Colorimetry has recently been devised for the assay of phenols based on nanozyme activity and received a lot of interest in the biological and environmental science due to its advantages of being simple and intuitive [22,23,24]. Different NPs with polyphenol oxidase mimicking activity are studied in the sensing applications. Ferrite NPs have piqued the interest of researchers due to their usage in bio-applications. They are ceramic materials made by physical and chemical blending of iron (III) oxide with modest amounts of other elements. The size, chemical composition, and particle interaction with the surrounding matrix all influence the properties of ferrite NPs [25,26,27]. To identify polyphenols, we have proposed a facile colorimetric method based on the polyphenol oxidase-mimicking activity of MgFe_2_O_4_@Mn_3_O_4_ nanocomposite. Catechol and resorcinol were taken as model polyphenols as they are commonly used and are easily available. Furthermore, in the quantitative colorimetric detection of catechol and resorcinol, the newly devised approach exhibits good sensitivity. This method, which is based on PPO-like activity of nanocomposite MgFe_2_O_4_@Mn_3_O_4_, shows tremendous promise for monitoring phenols in the environment. The research was extended to application of synthesized nanocomposite for colorimetric detection of polyphenols in the tea samples in order to assess the antioxidant effectiveness of foods rich in phenolics and define their perspective potential [28]. MgFe_2_O_4_ NPs and Mn_3_O_4_ NPs were synthesized using the sol-gel method, whereas the nanocomposite was fabricated using the ultrasonication method. Sol-gel synthesis provides numerous benefits over traditional approaches for producing high-quality NPs with uniformity and purity. It operates at low temperatures and has ability to produce fine powders [26]. Ultrasonication approach helps in fine blending of the NPs in the nanocomposite. The synthesized NPs and nanocomposite were fully characterized and effect of various parameters on enzyme mimic activity of MgFe_2_O_4_@Mn_3_O_4_ nanocomposite. The present work highlights the PPO-mimic potential of synthesized nanocomposite for detecting polyphenols.

## 2. Experimental

### 2.1. Materials and Reagents

All of the chemicals utilized were of analytical quality, viz. magnesium nitrate hexahydrate (Mg(NO_3_)_2_.6H_2_O), ferric nitrate nonahydrate (Fe(NO_3_)_3_.9H_2_O), citric acid monohydrate (C_6_H_8_O_7_H_2_O), ammonium hydroxide (NH_4_OH), manganese nitrate tetrahydrate (Mn(NO_3_)_2_.4H_2_O), catechol (C_6_H_6_O_2_), and resorcinol (C_6_H_6_O_2_). Deionized water was used for preparing the solutions.

### 2.2. Synthesis of MgFe_2_O_4_ NPs, Mn_3_O_4_ NPs, and MgFe_2_O_4_@Mn_3_O_4_ NC and Its Characterization

MgFe_2_O_4_ NPs were synthesized using sol-gel method where citric acid is used as a chelating agent (Figure 1) [29]. In 2:1 molar ratio, ferric nitrate (2.0 mol) and magnesium nitrate (1.0 mol) were dissolved in deionized water, followed by the addition of 4.5 mol citric acid to the reaction mixture. By adding ammonium hydroxide, the pH of the solution was neutralized, and the solution was turned into gel after continuous stirring at 60 °C for 4 h. The gel was dried at 160 °C and calcined for 3 h at 300 °C. These NPs were labelled as H-1 NPs. 

Sol-gel method was used to synthesize Mn_3_O_4_ NPs. In 20 mL distilled water, Mn(NO_3_)_2_ (0.09 mol) and citric acid (0.05 mol) were dissolved (Figure 2). The aforementioned mixture was magnetically agitated at 60 °C, and 20 mL of ammonium hydroxide (30% NH_4_OH) was added to neutralize it. The sol was converted into gel after 12 h of stirring. The gel was dried at 100 °C for overnight with increased volume. The gel was calcined at 450 °C for 3 h to acquire Mn_3_O_4_ NPs. These NPs were designated as H-2 NPs 

The nanocomposites (NC) of MgFe_2_O_4_ and Mn_3_O_4_ with *w*:*w* ratio of MgFe_2_O_4_:Mn_3_O_4_, i.e., 2:1 and 1:2 was synthesized by facile ultrasonication method. The synthesized MgFe_2_O_4_ and Mn_3_O_4_ NPs were sonicated separately in deionized water for 20 min. These two solutions were mixed and sonicated for 1 h which were designated as H-3 and H-4 NCs, respectively. H-4 NC displayed the best enzyme mimic activity as compared to H-3 NC and pristine H-1 and H-2 NPs, so rest of the experiments were performed using H-4 NC. Appendix A**,** lists the characterization tools used to evaluate surface morphology, particle size, phase purity, functional groups, and magnetic properties.

### 2.3. PPO-Mimic Activity and Kinetic Analysis

Zauberman method was used to assess the PPO-like activity of synthesized NPs. PPO catalyzes the oxidation of polyphenols to their respective o-quinones [27,30]. Nanocomposite was used instead of PPO utilizing catechol and resorcinol as substrates. 1 mg of synthesized NPs and 3.5 mL of 3 mM catechol/resorcinol solution at pH 3.0 were added to the reaction mixture. Activity of nanocomposite was assessed under different conditions of pH (1.0–7.0), temperature (10–50 °C), substrate concentration (1–5 mM), and enzyme mimic dosage (1–7 mg). The contact time was investigated upto 40 min. Each set of experimental data was optimized using the Box-Behnken methodology using design expert software (version 11).

The reaction kinetics was studied at 25 °C. The quantity of catechol and resorcinol was increased from 1 mM to 6 mM, and 5 mg of nanocomposite was added at an optimal pH of 3.0 with the addition of 0.5 mL H_2_O_2_ into each solution. The absorbance at 390 nm was measured at 1-min intervals for 5 min. From the Lineweaver-Burk plot, the Michaelis-Menton constant was determined as follows:
$$\frac{1}{V}=\frac{K_{m}}{V_{max}\left[S\right]}+\frac{1}{V_{max}}----\left(i\right)$$

where
“*V*” stands for reaction rate, *V_max_* for maximum velocity of reaction, *S* is the substrate concentration, and*K_m_* stands for Apparent Michaelis constant, which indicates the affinity of enzyme or enzyme mimic for its substrate. Lower value of *K_m_* corresponds to higher efficiency of catalyst.


### 2.4. Box-Behnken Approach for Atatistical Analysis

The current study used the Box-Behnken experimental design approach to investigate and standardise the following factors on enzyme activity: pH (A); Temperature (B); Catalyst dosage (C); Contact time (D). The Box-Behnken experimental design method establishes a mathematical link between components and experimental data, which can subsequently be fitted to a second-order polynomial model as the equation. As a result, Equation (1) is proposed for creating a mathematical correlation between enzyme activity and the four independent parameters chosen [31]. This equation is particularly useful in analyzing the impact of four selected experimental factors on the enzyme activity of MgFe_2_O_4_@Mn_3_O_4_ (1:2) nanocomposite and determining the optimum values.
Absorbance = β_0_ + β_1_A + β_2_B + β_3_C + β_4_D + β_5_AB + β_6_AC + β_7_AD + β_8_BC + β_9_BD + β_10_CD + β_11_A^2^ + β_12_B^2^ + β_13_C^2^ + β_14_D^2^(1)
where β_0_, β_ii_, β_ij_ are the model’s constant, linear, quadratic, and interaction coefficients. The Design-Expert Software was used to determine these constants (Version XI). Notably, the suggested model’s quality and goodness Equation (1), as well as the coefficient of determination (R^2^), were calculated. Aside from that, the residuals’ normal distribution was employed, as well as a plot of actual vs expected values. Analysis of variance (ANOVA) was used for statistical response analysis, with a *p*-value of 0.05 assumed to demonstrate the statistical significance of the proposed model’s parameters. Based on the experimental design, Appendix A (Given in the Appendix A) indicates the range and levels of the components.

### 2.5. Detection of Catechol and Resorcinol Based on MgFe_2_O_4_@Mn_3_O_4_ (1:2) Polyphenol Oxidase

Catechol detection was accomplished as follows:

100 µL of MgFe_2_O_4_@Mn_3_O_4_ (0.25 mg mL ^−1^) solution was mixed with various concentrations of catechol (0.1–0.8 mM) solution at pH 3.0. The absorbance at 390 nm was measured after 10 min of incubation at 25 °C.

Resorcinol detection was accomplished as follows: 

100 µL of MgFe_2_O_4_@Mn_3_O_4_ (0.25 mg mL^−1^) solution was mixed with various concentrations of resorcinol (0.01–0.08 mM) solution at pH 3.0. The absorbance at 390 nm was measured after incubating the solution at 25 °C for 10 min.

### 2.6. Detection of Polyphenols in Tea Samples Based on MgFe_2_O_4_@Mn_3_O_4_ (1:2) Polyphenol Oxidase Mimic

To estimate total phenol content, the Folin-Ciocalteu method was employed, which entailed mixing 20 μL of adequately diluted green and black tea extract (or phenolic standard) with 20 μL of Folin-Ciocalteu reagent for 3 min. Then, to make a total capacity of 1000 μL, 400 μL of sodium carbonate (7.5 percent Na_2_CO_3_) and deionized water were added. The solution was stirred for 60 min at room temperature in the dark, and the total polyphenolic content was determined using a spectrophotometer. The total phenolic content measured by the Folin-Ciocalteu assay was statistically correlated with the results from the study of two different tea types using for green and black tea by using optimized conditions of MgFe_2_O_4_ @Mn_3_O_4_ NC. Gallic acid (concentration range 20–100 μg/mL) was used as a standard for preparing calibration curve (Appendix A). Take different concentration of gallic acid followed by the addition of 5 mg of synthesized NC and 0.5 mL of H_2_O_2_ at pH = 3. Total volume was made 10 mL using distilled water. The value of absorbance corresponding to each test tube was noted to plot the calibration curve. The sample was prepared by adding 0.2 mL green and black tea extracts followed by the addition of 5 mg of NC and 0.5 mL H_2_O_2_ at pH = 3. The total volume of test sample was made 10 mL with the help of distilled water and total phenolic content was determined.

### 2.7. Recyclability Test

For the PPO-mimic reusability, 20 mL, 2 mM substrate at optimal pH of 3.0 was taken. 5 mg synthesized nanocomposite was added in the solution and the absorbance was measured at 390 nm after 10 min of incubation at 25 °C. The nanocomposite was then rinsed many times with deionized water. For the second cycle, the nanocomposite was centrifuged and dried in oven at 100 °C. This cycle was repeated four times.

## 3. Results and Discussions

### 3.1. Characterization of Synthesized NPs

#### 3.1.1. Structural Analysis

XRD was used to identify the phase and purity of synthesized NPs and nanocomposites. X-ray diffractograms displayed well-defined peaks (Figure 1), and the XRD parameters are listed in Table 1. The XRD pattern of H-1 NPs exhibited peaks at 2θ = 30.36°, 35.70°, 43.25°, 53.74°, 57.27°, 62.80°, 71.27°, and 74.38° [32,33]. These planes associated with the peaks proved to be strong evidence of the successful synthesis of MgFe_2_O_4_ (ASTM Data card No.17-465). Moreover, the absence of the extra peaks in the pattern made it evident that MgFe_2_O_4_ NPs were in a pure phase. The diffraction pattern of H-2 NPs showed sharp peaks at 2θ = 18.03°, 28.96°, 32.38°, 36.11°, 44.42°, 50.82°, 53.77°, 58.55°, 59.98°, and 64.67°, confirming the synthesis of Mn_3_O_4_ NPs (JCPDS Card No.24-0734). The Scherrer formula was used to recalculate the crystallite size (Table 1) of MgFe_2_O_4_ and Mn_3_O_4_, and their corrected values were 32 nm and 15 nm, respectively. The sharp diffraction peaks revealed that the NPs were well crystalline in nature. The combined peaks of H-1 and H-2 NPs were seen in the XRD patterns of H-3 and H-4, thus confirming their presence in the nanocomposite.

Fourier Transformation Infrared (FT-IR) was employed for studying functionalities in the synthesized NPs as illustrated in Appendix A. In case of H-1 NPs the band at 3424 cm^−1^ was assigned to O-H stretching and the band at 1633 cm^−1^ was accredited to O-H bending vibrations. Strong bands at 578 cm^−1^ and 480 cm^−1^ were observed confirming the formation of spinel structure. In addition, the band at 578 cm^−1^ were ascribed to the metal-oxygen bond vibrations in the tetrahedral lattice sites. Meanwhile, bands observed at lower frequency were assigned to metal-oxygen bond vibrations in the octahedral sites. The greater mass of Mn^3+^ ions in Mn_3_O_4_ resulted in higher band frequency as compared to MgFe_2_O_4_. The FT-IR spectrum of H-2 NPs showed a broad band at 3400 cm^−1^ and a narrow band at 1631 cm^−1^ due to stretching and bending vibrations of adsorbed water molecules. The small noticeable bands at 1384 cm^−1^ can be due to the presence of residues of nitrate whereas band at 1121cm^−1^ can be ascribed to C-OH stretching [34]. The band at 606 cm^−1^ was created by Mn-O stretching modes at tetrahedral sites, while the band at 487 cm^−1^ was caused by Mn-O distortion vibration in an octahedral environment [35,36,37]. The vibration of manganese species (Mn^3+^) in the octahedral sites caused the third band to be positioned at a weaker wave number 422 cm^−1^. The peaks of H-1 and H-2 NPs were observed in the spectra of H-3 and H-4 nanocomposites, confirming the presence of nanocomposite.

#### 3.1.2. Morphological and Magnetic Studies

SEM image of H-1 NPs (Appendix A) showed voids and pores as a result of gases released during gel combustion. Due to their magnetic nature and binding of primary particles via weak surface interactions such van der waals forces, ferrite NPs displayed a substantial degree of agglomeration. The presence of oxygen, iron and magnesium elements with compositions of 56.44%, 29.63%, and 13.94%, respectively, was observed in the EDS spectrum of H-1 NPs. Appendix A display SEM image of H-2 NPs which looks similar to irregular three-dimensional distorted nano-spheres. In addition, EDS recorded for Mn_3_O_4_ confirmed the presence of manganese and oxygen with composition of 72.79% and 24.76%, respectively. In case of H-3 and H-4 nanocomposites, (Appendix A and Figure 2) with the immobilization of NPs, the surface had a rough appearance, indicating low aggregation and consequently excellent dispersion. In addition, EDS of H-3 and H-4 NCs confirmed the presence of magnesium, manganese, iron and oxygen [38].

TEM imaging (Appendix A) was used to study the average particle size and aggregation behaviour of the pristine MgFe_2_O_4_ and Mn_3_O_4_ NPs. Distribution of size and morphology of the synthesized NPs was specified employing TEM analysis. MgFe_2_O_4_ NPs exhibited agglomeration in TEM image which is typical characteristic of magnetic NPs. The highest proportion of particles had a size in the range of 40–50 nm. TEM of Mn_3_O_4_ NPs shows spherical shaped particles having particle size in the range between 10–20 nm which was similar to as found through XRD studies [23].

Hysteresis plots displaying the altering of magnetization (M_s_, emug^−1^) as an effect of applied magnetic field (H, O_e_) is illustrated in Figure 3. The magnetic character of the H-1 NPs depends crucially on the size, shape and purity of the NPs. It is very interesting that MgFe_2_O_4_ shows magnetism even though Mg^2+^ ions are non-magnetic. This can be ascribed to the incomplete inverse spinel structure of MgFe_2_O_4_.The H-2 NPs showed linear behaviour in M−H curves, indicating their paramagnetic character at 300 K as shown in Figure 3. Hysteresis plot in single-domain ferrimagnetic particles vanishes when particle size becomes so small that the maximum anisotropy energy becomes close to the thermal energy, and the process of flipping of the single domain spin becomes inhibited. These types of particles do not possess any coercivity and their magnetization never becomes saturated even at very high applied field [26,36,37]. The presence of hysteresis loop indicates that particles are multidomain in the present studies. At room temperature, the ferrite NPs displayed s-shaped thin hysteresis loop, indicating their ferrimagnetic nature. The M_s_ values for H-1, H-2, H-3 and H-4 were 23.34 emug^−1^, 0.86 emug^−1^, 19.53 emug^−1^, 18.48 emug^−1^ and 18.34 emug^−1^. The saturation magnetization values of H-3 NC decreased in comparison to pure H-1 NPs. This fall in M_s_ value was assigned to the presence of non-magnetic Mn_3_O_4_ in H-3 and H-4 NC which influences the magnetization as a result of quenching magnetic momentum. The surface imperfections of the ferrite NPs also contribute to the drop in M_s_ values. The values of coercivity drop as the presence of manganese ion in H-3 and H-4 increases. The other magnetic parameters, i.e., the remanence magnetization (M_r_) also falls with the addition of nonmagnetic Mn^3+^ ions (Table 2). The narrow loops hinted the low coercivity values which implies that the synthesized ferrites could be demagnetized easily.

The BET surface area of H-1, H-2, H-3 and H-4 NPs was recorded as 32.81 m^2^g^−1^, 38.98 m^2^g^−1^, 53.03 m^2^g^−1^, and 70.17 m^2^g^−1^, respectively. The total pore volume of H-1, H-2, H-3 and H-4 was 0.030 ccg^−1^, 0.122 ccg^−1^, 0.17 ccg^−1^, and 0.18 ccg^−1^, respectively. According to the IUPAC classification, representative of mesoporous structures, all samples exhibited typical type-IV isotherms with hysteresis loop of H4 type and therefore depicting the slit shaped pores. The BJH pore size distribution displayed one narrow peak centred at 3.59 nm and 6.59 nm for H-1 and H-2 NPs, respectively. H-3 and H-4 exhibited the unimodal pore size distribution as well displaying peaks centred at 11.32 nm and 16.47 nm, respectively [39]. The H-3 and H-4 NC exhibited higher surface area as well as pore volumes explaining the enzyme mimic activity. With increasing Mn_3_O_4_ content the surface area increased from 53.03 m^2^g^−1^ for H-3 NC to 70.17 m^2^g^−1^ for H-4 NC (Figure 4 and Table 3). 

### 3.2. PPO-Mimic Activity

Due to the presence of MgFe_2_O_4_ the surface area of nanocomposite increased which resulted in greater interaction with substrate causing enhanced enzyme mimic activity. Since PPO-like activity is also dependent on electron transfer, the substrates transfer their electron to NPs by electron transfer to form catecholate/resorcinolate, which subsequently generates o-quinones and 1,3-Benzoquinone via electron transfer in the presence of dissolved O_2_ (Figure 3 and Appendix A). Among NPs and nanocomposite, H-4 NC displayed the highest activity followed by H-3 and H-2 (Appendix A). Lowest activity was shown by H-1 NPs. These results indicate that in the nanocomposite Mn_3_O_4_ is the catalytic centre. These results revealed that at lower MgFe_2_O_4_ concentration as in H-4 NC the activity was increased due to lesser agglomeration of NPs. PPO activity also depends upon electron transfer process and surface area but electron transfer process was dominated. Without electron transfer process, catechol cannot be reduced to catecholate further into semiquinone free radicals and o-quinone [40,41,42,43]. Mn^3+^ is expected to gain an electron in Mn_3_O_4_ (Mn^2+^O.Mn^3+^_2_O_3_) NPs through direct coordination of catecholate with metal into unoccupied sigma orbital (3d^4^ to 3d^5^), with no change in spin state during electron transfer. Mn^3+^ is reduced to Mn^2+^ by accepting an electron from catechol. The presence of MgFe_2_O_4_ increased the surface area of the nanocomposite as depicted in Figure 4.

Iron (Fe) has an electronic configuration of [Ar] 3d^6^4s^2^, but in MgFe_2_O_4_ NPs, iron is present as Fe^3+,^ which already has a stable electronic configuration of 3d^5^, thus it cannot take an electron from catechol and hence cannot participate in the electron transfer process, resulting in no reaction [44]. The activity of H-3 NC was lowered as compared to H-4 NC because higher content of MgFe_2_O_4_ hampered the electron transfer process. Catechol and resorcinol solutions were characterized by an λ_max_ peak at 285 nm and 274 nm. The oxidation of catechol and resorcinol in the presence nanocomposite was determined spectrophotometrically by the formation of o-quinones which showed absorbance at 390 nm. Increase in absorbance directly correlated with the PPO-mimic activity. Higher the absorbance, more is the PPO-like activity. As H-4 NC showed the best activity among all the synthesized samples. Further experiments were performed using this nanocomposite.

Different parameters impacting enzyme mimic activity were studied, with the findings displayed in Figure 5. Figure 5a shows the effect of changing the pH of the solution from 1.0 to 9.0 on nanozyme activity. At pH 1.0, the nanocomposite activity peaked. After pH 3.0, the activity dropped dramatically. The stability of semiquinone free radicals at high pH was the cause of the reduction in activity. However, in order to avoid the acidic conditions pH 3.0 was optimized for further experiments. Temperature had an effect on activity, as shown in Figure 5b, with the maximum activity being recorded at 25 °C and the reaction rate was decreased when temperature was increased, indicating that 25 °C was the ideal temperature. Since less catecholate/resorcinolate was oxidized to semiquinone free radicals as the temperature increased, the formation of surface complexes between catechol/resorcinol and nanocomposite reduced. Similarly, the concentration of substrate and the dose of NC were investigated (Figure 5c,d). The optimal substrate dosage was observed to be 2 mg. There were less semiquinone radicals at extremely low concentrations of catechol and resorcinol, resulting in decreased activity, but catechol/resorcinol at higher concentrations has inadequate capacity to completely oxidize to semiquinone radicals, resulting in build-up of semiquinone radicals. The most effective dose was found to be 5 mg. When the dose was increased up to 5 mg, the activity of nanocomposite increased, but when the concentration of nanocomposite exceeded the ideal limit, it caused diminished activity. 10 min was found to be the ideal contact time (Figure 5e). Since as the time of contact increased, a greater number of NPs interacted with the substrate, the activity of the NC increased, and thus the absorbance increased. Curve flattened after reaching its peak at 10 min, and no more increase is observed after this point [45]. 

### 3.3. Kinetic Studies of PPO-Like Activity and Recyclability Test

The variation in the absorbance of the reaction at 390 nm was monitored for 5 min at 1 min intervals for varied dosages of catechol and resorcinol. The change in catechol and resorcinol absorbance at varied concentrations of nanocomposite is shown in Figure 6a,b. The rate constants were determined for various concentrations using log [A_o_/(A_o_—-A_t_)], where A_o_ and A_t_ indicate the absorbance levels at infinite time and time “t”, respectively. The reaction of nanocomposite followed first-order kinetics at low substrate concentrations and zero order kinetics at higher substrate concentrations, as shown in Appendix A. The kinetic parameters *K_m_* and *V_max_* were calculated from the Lineweaver-Burk plot shown in Appendix A. *K_m_* is an indicator of the enzyme affinity to the substrate. As nanocomposite has lower value of *K_m_* (app) = 0.7 mM as compared to Mn_3_O_4_ (*K_m_* (app) = 1.14 mM) therefore, nanocomposite has higher enzyme affinity to bind to the substrate. Presence of Mn_3_O_4_ facilitated electron transfer process and presence of MgFe_2_O_4_ increased the surface area of the NC. As a result, a decreased *K_m_* value indicated that synthesized NC as effective PPO mimics.

Figure 7 shows the reusability result, which shows that after four cycles, 79 percent of the PPO-mimicking activity was preserved. The loss of sample amount throughout each cycle resulted in a drop in mimic activity. CTAB@MgFe_2_O_4_ nanocomposite as peroxidase mimic were found to have a comparable drop in activity by Singh et al., 2021 [24]. After six months of storage at room temperature, the catalytic activity of the synthesized nanocomposite remained unaltered, showing the nanocomposite persistence as PPO-mimic.

### 3.4. Mechanism of PPO Mimic Activity

Naturally occurring PPO contains copper and the enzyme is involved in catalyzing the oxidation of polyphenolic compounds. The ability of nanocomposite to act as a PPO-mimic by catalyzing the oxidation of the usual PPO substrate catechol and resorcinol was proven in this study. In comparison to natural PPO, nanocomposite showed superior stability. Catechol and resorcinol release their hydroxyl hydrogen to the PPO-mimic, forming catecholate and resorcinolate, which consequently result in formation of semiquinone free radicals via electron transfer. In the case of nanocomposite, electron gain by Mn^3+^(catalytic centre) was most likely accomplished via direct coordination of catecholate with metal, with no change in spin state. Mn^3+^ is reduced into Mn^2+^ by receiving an electron from catecholate/resorcinolate. The role of MgFe_2_O_4_ is to increase the surface area of the nanocomposite which enhanced activity of nanocomposite as compared to pristine Mn_3_O_4_ NPs. In addition to the contact and interaction between enzyme mimic and substrates, electron transport occurs in the catalytic core, which is a significant element in controlling the reaction rate [27]. This study is the first look at the enzyme mimicking activity of MgFe_2_O_4_@Mn_3_O_4_ nanocomposite.

### 3.5. The Statistical Analysis Findings

The Box-Behnken statistical model was employed using the detailed input data listed in Table 4. (Four independent variables and 30 runs). This table shows the actual and expected catechol and resorcinol concentrations, as well as the experimental design matrix for the Box-Behnken statistical model. As a result, a quadratic model was developed to estimate enzyme activity of nanocomposite using catechol and resorcinol extracted from aqueous solution. The quadratic models for catechol and resorcinol are shown in Appendix A, respectively. This model can be written as shown in Equations (2) and (3) with 4 coded factors for catechol and resorcinol, respectively, as:Absorbance (Using catechol) = 2.05 − 0.1167A − 0.0167B + 0.000C − 0.1167D + 0.0500AD + 0.2750BD + 0.2375C^2^ − 0.2625D^2^(2)
Absorbance (Using resorcinol) = 2.17 + 0.1250 A − 0.0583 B − 0.0667 C − 0.0333D + 0.4250 AB + 0.4250CD(3)

The normal probability versus plot of residuals from the least-squares fitting for the response catechol and resorcinol, is graphically depicted in Appendix A. The points on the plot are also quite near to a straight line since the residuals follow a normal distribution pattern. In Appendix A, a random scatter plot of the actual values and expected findings of catechol and resorcinol is shown. Both the expected and actual results are distributed randomly around the 45° straight line, suggesting that the error values between the actual and anticipated values have a zero mean. All of these findings point to the proposed model’s strong correlation and appropriateness in predicting enzyme mimic activity using catechol and resorcinol. 

### 3.6. Standardization of the Experimental Parameters

The optimization procedure’s purpose is to find a set of experimental variable levels where enzyme activity (as measured by catechol and resorcinol) is at its peak. This is accomplished using the Box-Behnken design process, which is an excellent technique for determining the optimal operating conditions among a range of experimental variables evaluated. The results showed that the suggested model properly predicted the best enzyme activity utilizing catechol and resorcinol under optimal conditions, which included pH = 1.0; temperature = 25 °C; catalyst dose = 5 mg; and contact time = 10 min. Enzyme mimic was studied activity using catechol as substrate at pH ranging from 1–9, with variation in contact time = 2–20 min (Figure 8a).and using resorcinol as substrate at temperature ranging from 10–60 °C, with variation in pH (Figure 8b). The factor of desirability was 1 which represents a desirable, or ideal response (Figure 9). These graphs show that as the contact time grows up to a certain point, strong enzyme mimic activity occurs at acidic pH values [46]. Another crucial aspect to investigate is the dose of catalyst nanoparticles, as it may have a substantial impact on the performance of enzyme mimic activity. Furthermore, from an economical perspective, determining the optimal catalyst quantity is one of the most important attributes. The enzyme mimic activity increased with increasing NP dosage and contact time up to a specific limit, as shown in Figure 8b. An increase in the amount of PPO-mimic in the solution causes this behaviour, which increases the surface area or reaction sites.

### 3.7. Colorimetric Detection of Catechol and Resorcinol

Catechol and resorcinol were determined using a sensitive and straightforward approach. The variations in absorbance spectra in the presence of catechol at various concentrations shown in Figure 10a and Figure 11a (0.1 mM to 0.8 mM) at 25 °C at 390 nm using H-4, i.e., MgFe_2_O_4_@Mn_3_O_4_ (1:2) nanocomposite solution of pH 3.0 with the addition of 0.5 mL H_2_O_2_ into respective solutions. With increasing catechol content, the absorbance increased in a consistent manner. The linear connection between catechol concentration and system absorption intensity is shown in Appendix A, with both proportionate to each other in the range of 0.1–0.8 mM (R^2^ = 0.986) and a detection limit of 0.29. This technique, as anticipated, might also be utilized to identify resorcinol. The absorbance spectra variations in the presence of resorcinol (0.01 to 0.08 mM) using H-4 NC solution having temperature 25 °C with pH 3.0 at 390 nm are represented in Figure 10b and Figure 11b. It’s clear that when the concentration of resorcinol rises, so does the absorbance of the solution. The relationship between the strength of absorption and the concentration of resorcinol in the solution is shown in Appendix A, with both being proportionate in the range of 0.01–0.08 mM (R^2^ = 0.926) and a detection limit of 0.03. Table 5 summarizes the efficacy of different PPO- mimic for catechol and resorcinol detection. The other polyphenol oxidase biosensors include electrochemical sensors using PEDOT-Gr/Ta electrode for the measurement of hydroquinone, catechol, resorcinol and nitrite [47]. Likewise, fluorescence-based sensing using titanium dioxide (P-TiO_2_) nanoparticles (NPs) and fluorescein, has been used for sensing of catechol [48]. The novelty of present work is colorimetric sensing of polyphenols. The role of MgFe_2_O_4_@Mn_3_O_4_ nanocomposite is that it catalyzes the oxidation of catechol and resorcinol to semiquinones and gives faint brownish colour depending upon the concentration of polyphenol which is the basis for colorimetric sensing application. The colorimetric biosensor showed a stable increase in colour intensity with increasing catechol and resorcinol concentration at pH =3.0. However, it is more sensitive for resorcinol showing faint pink to brownish colour and the biosensor gives light brown colour for catechol. The colorimetric biosensor using MgFe_2_O_4_@Mn_3_O_4_ nanocomposite has advantage of being simple, portable and easy to use.

### 3.8. Colorimetric Sensing of Polyphenols in Tea Samples

Two samples of green and black tea were used in the assay, and the results were compared to those obtained using traditional methods. The colorimetric sensing of polyphenols in tea samples in the presence of MgFe_2_O_4_@Mn_3_O_4_ in the wavelength ranging from 400–700 nm was used to determine the total phenolic content of aqueous green and black tea extracts. The total phenolic content (R^2^ = 0.0995) measured by the Folin-Ciocalteu assay, which comes out to be 62.2 mg/100 mL and 44.5 mg/100 mL, has a statistically significant linear association with the results from the study of two different tea types using nanocomposite as 63.3 mg/100 mL and 45.3 mg/100 mL for green tea and black tea, respectively (Figure 12). The role of MgFe_2_O_4_@Mn_3_O_4_ (1:2) nanocomposite is that it catalyzes the oxidation of polyphenols which are present in green and black tea extracts into quinones [28,51,52]. In order to ascertain the complex phenolic composition of tea such as catechin, esculetin, caffeic acid and rutin the advanced analytical approaches consisting of both LC-LTQ-Orbitrap Fourier transformed (FT)-MS and LC-time-of-flight-(TOF)-MS coupled to solid-phase extraction (SPE) NMR have been reported [53].The elaborated chromatographic method enabled separation, qualitative and quantitative determination of five catechins (C, EC, EGC, EGCG and ECG) and gallic acid, the main phenolic acid, which esterifies polyphenols in tea [54]. However, the present study was focussed on facile estimation of total phenolic content of tea samples by colorimetric method using synthesized MgFe_2_O_4_@Mn_3_O_4_ nanocomposite as PPO-mimic. 

## 4. Conclusions

The MgFe_2_O_4_@Mn_3_O_4_ nanocomposite-based platform could be used to detect catechol and resorcinol in the range of 0.1–0.8 mM and 0.01–0.08 mM, respectively, under optimized conditions, with detection limits of 0.20 mM for catechol and 0.03 Mm for resorcinol. The findings of the tea variety study show a statistically significant linear correlation with total phenolic content (R^2^ = 0.0995) as measured by the Folin-Ciocalteu test. These findings indicate that MgFe_2_O_4_@Mn_3_O_4_ nanocomposite with optimized ratio can be used for colorimetric sensing of polyphenols. Furthermore, MgFe_2_O_4_@Mn_3_O_4_ nanocomposite as PPO mimics shown several advantages over natural PPO enzyme, including ease of synthesis, storage, and stability. The nanocomposite can be employed as PPO mimics in a variety of applications, including biosensing and polyphenolics removal in industrial waste waters. The results afford platform for potential applications of the developed system in different fields such as clinical diagnosis and environmental monitoring of polyphenols.

## Data Availability

Supporting data available.

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
