# Peer review of "Nanocomposite of MgFe2O4 and Mn3O4 as Polyphenol Oxidase Mimic for Sensing of Polyphenols"

_biosensors, 2022, doi:10.3390/bios12060428_

Round 1

Reviewer 1 Report

Dear editor,

The manuscript “Nanocomposite of MgFe2O4 and Mn3O4 as polyphenol oxidase

mimic for sensing of polyphenols”. The results are promisor for the scientific community of this area. Indeed, the authors used statistical tools that have been demonstrated the greatness of the work. Therefore, I recommend the publication of this work in Biosensors with some modifications. My decision is leveraged by some questions and suggestions to improve the work for a review, as I describe below.

ABSTRACT

1.     The abstract must be improved. In my perspective the text is badly described.

INTRODUCTION

2.     In lines 45-49 the authors comment about the methods for detecting and determinate polyphenols. The authors forgot to mention the electrochemical methods using enzymes and mimetics. I recommend to cite some articles that are well established in the literature about determination of phenolic compounds in food matrices, as the mentioned work. Some examples of references: (10.1016/j.mtcomm.2020.101251; 10.1016/j.mtcomm.2022.103481; 10.1016/j.talanta.2009.01.038; 10.1016/j.talanta.2008.07.003).

3.     It was not clear for me why the authors have been commented in the introduction section about the hazardous water pollutant of phenols, but the determination of catechol and resorcinol is in tea samples. I think the authors must change the perspective of the Introduction or added a paragraph writing about the phenolic content in food matrices, especially in tea samples.

EXPERIMENTAL SECTION

4.     The authors must provide a Scheme of synthesis to better understand the sol-gel synthesis of MgFe2O4 and Mn3O4 NPs. (section 2.2)

5.      In section 2.3 the authors made the kinetics of studied reactions. The evaluated Km is not correct to designate. The correct form is Km,app. (Michaelis-Menten apparent). The true constant values are independent of concentration of reactants, activators, inhibitors, extraneous agents and pH. The apparent rate constants are such constants of the composite reaction which are observed when this reaction is described by the equation of simple reaction. Michaelis constant calculated by a half of the ultimate constant is an apparent constant. The apparent constants may be functions of several true rate constants and/or concentrations of reacting substances. The evident physical sense of apparent constants being absent, only formal relation between the reaction rate and reactant concentration independent of the investigated mechanism is provided. (see more in: On true and apparent Michaelis constants in enzymology. I. Differences]. Ukr Biokhim Zh (1999). 2011 Sep-Oct;83(5):94-109)

RESULTS AND DISCUSSION

6.    The characterization of materials is well described. I just have some doubts about FTIR attributions. Why are there some O-H stretching’s? Water of residue of sol-gel synthesis? The attributions of Mn-O must be referenced. Indeed, I don’t believe in C-OH vibrations in 1384 and 1121 cm-1.

7.     The discussion of magnetic hysteresis must be improved. I think that the discussion of different proportions of MgFe2O4 and Mn3O4 can give a great discussion.

8.     In section 3.3 why the authors explain the Km(app) value of 0.7 mM compared to Mn3O4 (Km(app) = 1.14 mM). I think that a chemical explanation must be present in this section.

9.     Why the authors choose catechol and resorcinol?

10.  The authors must add a section of interferents studies with other phenolic compounds found in tea samples such as catechin, esculetin, caffeic acid, rutin.

Author Response

Respected Reviewer

Thanks for the suggestions

The replies are attached as word files

Best regards

Reviewer 2 Report

Manuscripts reports under the title “Nanocomposite of MgFe2O4 and Mn3O4 as polyphenol oxidase mimic for sensing of polyphenols” The content of the work is interesting, but the manuscript cannot be published in the present form due to the following issues:

1.      The description of the introduction section should enhance. The sol-gel method has been used for the preparation of MgFe2O4 NPs, and Mn3O4 NPs whereas the ultrasonic method has been used for the preparation of the composite. The reasoning including the advantages of the different methods should be included in the last paragraph of the introduction.

2.      The Wt % of the element is required for every EDS image. Please put the data in the form of a table as an inset of the figures.

3.      Grammar and lots of typological errors are present in the present form of the manuscript. So need an extensive rectification.

4.      The TEM of (a) H-1 NPs and (b) H-2 NPs of Figure S3 needs to show the interplanar spacing of the respective elements.

5.       Author should involve the novelty of the work with the other polyphenol oxidase biosensors. This should be added before the conclusion part

Author Response

Respected reviewer

Thanks for the comments and suggestions that have helped in improving the manuscript

The word file is attached herewith

Best regards

Round 2

Reviewer 1 Report

The manuscript was improved and all suggestions and doubts were resolved. Therefore, I recommend publishing this manuscript in Biosensors.

Reviewer 2 Report

The revised manuscript is acceptable for publication.